# Unravelling the Signature Follicular Fluid Metabolites in Dairy Cattle Follicles Growing Under Negative Energy Balance: An In Vitro Approach

**DOI:** 10.3390/ijms252312629

**Published:** 2024-11-25

**Authors:** Muhammad Shahzad, Jianhua Cao, Hubdar Ali Kolachi, Jesse Oluwaseun Ayantoye, Zhou Yu, Yifan Niu, Pengcheng Wan, Xueming Zhao

**Affiliations:** 1Institute of Animal Sciences (IAS), Chinese Academy of Agricultural Sciences (CAAS), No. 2 Yuanmingyuan Western Road, Haidian District, Beijing 100193, China; drmshahzadvet@gmail.com (M.S.);; 2State Key Laboratory of Sheep Genetic Improvement and Healthy Breeding, Institute of Animal Husbandry and Veterinary Sciences, Xinjiang Academy of Agricultural and Reclamation Sciences, Shihezi 832000, China

**Keywords:** negative energy balance, follicular fluid, metabolites, dairy cattle

## Abstract

The astringent selection criteria for milk-oriented traits in dairy cattle have rendered these animals prone to various metabolic disorders. Postpartum lactational peak and reduced feed intake lead to negative energy balance in cattle. As a compensatory mechanism, cattle start mobilizing fat reserves to meet the energy demand for vital body functions. Consequently, diminished glucose concentrations and elevated ketone body levels lead to poor ovarian function. The impaired follicular development and subpar oocyte quality diminish the conception rates, which poses significant economic repercussions. Follicular fluid is integral to the processes of follicular growth and oocyte development. Hence, the present study was performed to identify potential alterations in metabolites in the follicular fluid under in vitro culture conditions mimicking negative energy balance. Our results revealed nine distinct metabolites exhibiting differential expression in follicular fluid under negative energy balance. The differentially expressed metabolites were predominantly associated with pathways related to amino acid metabolism, lipid metabolism, signal transduction mechanisms, and membrane transport, alongside other biological processes. The identified signature metabolites may be further validated to determine oocyte fitness subjected to in vitro fertilization or embryo production from slaughterhouse source ovaries.

## 1. Introduction

Intensive dairy farming has evolved significantly due to persistent technological innovations and selective breeding practices [1,2]. The milk output from dairy cattle has shown an increasing trend throughout the previous century and is expected to witness further enhancements [3]. Over the last forty years, milk production in the United States has risen two-fold, thus standing at six times the level recorded a century prior [4]. Nonetheless, the milk yield improvement has also been associated with a notable decline in fertility [5]. The extended postpartum anestrus results in an elongated calving interval, which engenders significant economic consequences [6]. The incidence of acyclicity concerning postpartum anestrus is estimated at nearly 26.3%, with specific instances surpassing 40% linked to metabolic dysfunctions in high-yielding cattle [7].

The postpartum phase is the most vulnerable period for metabolic disturbances in high-producer cows, coinciding with the onset of lactation. The early lactation stage is typically characterized as the first 100 days after calving [8]. During this phase, cows reach their highest milk production, with Holstein cows achieving this peak in the second month [9]. The absorbed nutrients are maximally utilized to facilitate the synthesis of milk in high producers [10]. In this way, a significant amount (85%) of circulatory glucose (Glu) is mobilized for lactose synthesis during lactogenesis [11]. Approximately, 2.7 kg of plasma Glu is extracted per day in lieu of 40 kg of milk by the mammary epithelial cells [12]. Nevertheless, insufficient rumen expansion and hormonal fluctuation restrict feed and water intake during the early postpartum phase [13]. Under these circumstances, high-yielding cows exhaust body reserves and undergo considerable weight loss (≥60%) to combat energy deficit [14]. Simultaneously, low feed consumption and depletion of reserves lead to a negative energy balance (NEB).

In postpartum cows, decreasing circulatory Glu levels during the NEB period trigger glucagon secretion, while insulin release is reduced [15,16]. Reduced insulin production inhibits Glu absorption in peripheral tissues that are sensitive to insulin, but increases it in mammary tissues that are insensitive to insulin [17]. As a compensatory mechanism, the NEB state stimulates the breakdown of triglycerides in body fat reserves and releases non-esterified fatty acids (NEFAs). The excessive NEFAs are processed by the liver and converted to ketone bodies (β-hydroxybutyric acid/BHBA) or re-esterified to triglycerides [18]. Meanwhile, skeletal muscle adapts to NEB by utilizing NEFA and ketone bodies as alternative energy sources [14]. The excess of ketone bodies in the bloodstream can lead to subclinical ketosis. The impact of subclinical ketosis is so pronounced that it affects nearly 24.3% of cattle worldwide [19]. Besides general circulation, ketone bodies are also transduced to the follicular fluid (FF) in the ovarian follicles [20].

Follicle development and ovulation depend on ovarian FF, which offers an optimal microenvironment for oocyte maturation [21,22]. FF is a rich source of numerous metabolites, hormones, amino acids, cytokines, growth factors, energy substrates, lipids, and cholesterols essential for oocyte maturation [23]. Interestingly, the biochemical profiles of FF and blood plasma have a remarkable resemblance [14]. Thus, any variation in the composition of plasma is likewise reflected in the FF [24]. Consequently, diminished Glu and augmented BHBA in FF have a deleterious effect on follicular development and oocyte quality in cattle [25].

Half of the postpartum ovarian issues experienced by dairy cows are caused by ovarian inefficiency, which significantly impairs the reproductive performance of the herd as a whole [26]. Oocyte quality has drawn much attention in the breeding sector, especially with the adoption of in vitro embryo production technology [27]. A thorough analysis of FF metabolites in connection to NEB may offer a profound understanding of the reasons behind the reduction in dairy cow fertility. High-throughput omics tools are hallmarks of revealing complicated biological processes [28]. Metabolomic studies have been performed to elucidate the physiology and pathology of ovarian FF in postpartum cattle [26]. The molecular characterization of metabolites may offer a more precise comprehension than traditional approaches [29,30]. Furthermore, differential expression of metabolites may reveal biomarkers and mechanisms connected to variations in follicular development and infertility [31].

Nevertheless, investigations into the relationships between NEB and the metabolomics of FF remain inadequate. Given this importance, the current study was undertaken to investigate the metabolic alterations in FF resulting from NEB-mimicking culture conditions. Using high-throughput non-targeted metabolomics techniques it was attempted to reveal differential metabolites of the FF under NEB.

## 2. Results

### 2.1. Test Data Quality Analysis

The comprehensive ion current profile of quality control (QC) samples was superimposed and analyzed to ascertain the fidelity of the data. The acquired data were considered credible, as the positive and negative ion spectra indicated that the response intensity and retention time (Rt) of each chromatographic peak coincided (Figure 1A,B).

### 2.2. Principal Component Analysis (PCA)

According to the findings derived from Pearson correlation analysis, the QC samples in both positive and negative ion modes exhibited a notable degree of clustering (Figure 2A,B). The correlation coefficient among the QC samples exceeded 0.9 (value closer to 1), indicating a high level of reproducibility in the experimental procedure and affirming the reliability of the subjected data (Figure 3A,B).

### 2.3. Identification and Classification of Metabolite

Table 1 enlists the combined statistics of the 1111 metabolites that were identified in total. As described by Yang et al., using information from the KEGG database, the identified metabolites were categorized based on their chemical classification [32]. According to the chemical classification, the metabolites were further categorized and quantified in their respective classes. Carboxylic acids and derivatives constituted the highest proportion at 19.62%. Unknown metabolites represented 14.40%; glycerophospholipids comprised 10.26%; fatty acyls accounted for 9.72% of the total metabolites. Organic oxygen compounds made up 6.48%; benzene and derivatives contributed 5.67%; pregnenol ketones represented 4.05%; sterols and derivatives accounted for 3.96%; organic nitrogen compounds constituted 2.43%; sphingosine lipids comprised 2.34%, among others (Figure 4).

### 2.4. Univariate Statistical Analysis

Univariate statistical analysis, including fold change (FC) analysis, and Student t-test/non-parametric assessments were applied. A comprehensive analysis was performed on a total of 1111 metabolites identified in both positive and negative ionization modes. The criteria for screening were established as FC > 1.5 or FC < 0.67, *p* < 0.05, and the metabolites identified as differentially expressed were illustrated through volcano plots (Figure 5A,B).

### 2.5. Screening for Differential Metabolites

Metabolomic analysis of FF revealed a significant difference between the metabolites of NEB and PEB groups. In the positive ion mode, 25 metabolites were differently expressed (Table 2). This includes the upregulation (↑) of 16-hydroxy palmitic acid (VIP = 1.61, *p* < 0.01) and downregulation (↓) of creatine (VIP = 10.08, *p* = 0.03). In negative ion mode, 39 metabolites were differentially expressed (Table 3). Notably, lipid-related compounds, like 1-oleoyl-sn-glycero-3-phosphocholine, were upregulated (VIP = 3.96, *p* < 0.00, ↑), whereas 1-palmitoyl-2-linoleoyl-sn-glycero-3-phosphate was downregulated (VIP = 1.46, *p* = 0.01, ↓). Amino acids and their derivatives expression also showed significant differences between the two groups. For instance, asparagine (VIP = 2.28, *p* = 0.01, ↓) and L-proline (VIP = 2.55, *p* = 0.02, ↓) were downregulated under the NEB condition. The 2′ deoxycytidine showed downregulation (VIP = 1.79, *p* = 0.02, ↓) in the NEB group as well. Overall, a significant alteration in metabolic profile was observed in the NEB follicles of the cattle.

### 2.6. Clustering Analysis

Differently expressed metabolites were illustrated in the form of heat maps. The visualization highlights the distinct marker metabolites between NEB and PEB groups. In positive ion mode, significant metabolites were identified. Elevated expression encompassed 16-hydroxy palmitic acid and L-alanine, whereas trimethylamine-N-oxide, 2′-deoxycytidine, and creatine exhibited notable reductions. (Figure 6A). Conversely, in negative ion mode, asparagine, palmitoleic acid, BHBA, and pantothenic acid exhibited intensified expression. In contrast, L-proline demonstrated a decreasing trend. This analysis facilitates the identification of metabolic alterations under the conditions studied (Figure 6B).

### 2.7. KEGG Pathway Enrichment Analysis

The KEGG (Kyoto Encyclopedia of Genes and Genomes) enrichment analysis conducted on the metabolites exhibiting significant alterations between the NEB group and PEB group reveals critical insights. The analysis indicates that pathways associated with the synthesis and degradation of ketone bodies, mineral absorption, and central carbon metabolism in oncological contexts were notably affected. The analysis also elucidated the involvement of amino acid biosynthesis, translational mechanisms, particularly aminoacyl-tRNA biosynthesis, nucleotide metabolism with a focus on pyrimidine metabolism, as well as pathways related to membrane transport, including ATP-binding cassette (ABC) transporters, and lipid metabolism, encompassing both the synthesis and degradation of ketone bodies. As depicted, the metabolic pathways related to amino acids, lipids, membrane transport, and nucleotides were identified as the predominant pathways in the FF of developing follicles under conditions of NEB (Figure 7A,B).

## 3. Discussion

Dairy cattle subfertility is a complex and multifaceted issue of global significance. Lactation induces a shift in the metabolic homeostasis of cattle [33]. Any metabolic discrepancy may further alter the follicular microenvironment resulting in poor fertility [26]. The application of molecular approaches utilizing untargeted omics techniques may yield detailed information regarding the metabolites contributing to this problem [34]. Our study encompasses the metabolic changes occurring during the NEB state in dairy cattle. Using untargeted metabolomics, we found 26 different metabolites in FF between the NEB and PEB groups in this study. Of these, 10 were annotated through the KEGG pathway, and five positive differential metabolites were screened, including 16-hydroxy palmitic acid, 2′-deoxycytidine, L-alanine, creatine, and trimethylamine N-oxide.

16-hydroxy palmitic acid, a hydroxylated fatty acid, is crucial to the oxidative metabolism of fatty acids in mammalian tissues. This is a vital biochemical process for maintaining cellular energy equilibrium, particularly in states of starvation [35]. It has been demonstrated that polyunsaturated fatty acids possess the capacity to regulate the immune system and the inflammatory response [36,37]. A strong positive correlation exists between the inflammatory mediators (interleukin-6 and interleukin-8) and 16-hydroxy palmitic acid [38]. In the present investigation, 16-hydroxy palmitic acid was upregulated in the FF. Our finding reflects that the follicles under NEB conditions could exhibit heightened susceptibility to inflammatory agents. An in vitro study showed that interleukin-6 may deteriorate the oocyte quality by disrupting the spindle and chromosomal structure [39]. The elevated interleukin-6 may cause ovarian cyst formation and poor fertility in cows [40]. Gross inflammation leads to ovarian ageing and influences follicular development [41].

Deoxycytidine (2′-deoxycytidine) is essential in synthesizing deoxyribonucleic acid (DNA). Consequently, it is instrumental in translation and transcription processes that pertain to cellular metabolism [42]. Thus, through this critical role, it contributes to the regulation of cellular growth, proliferation, differentiation, and inhibition [26]. Our observations indicated that 2′-deoxycytidine was significantly enriched in nucleotide metabolism and membrane transport pathways (KEGG analysis). Moreover, the expression of 2′-deoxycytidine in FF was markedly downregulated in the NEB group, given its pivotal role in the nucleotide metabolism pathway, which is crucial for cellular development and metabolic processes [43]. Downregulation reflects a compromised follicular function. However, insufficient information is available regarding the role of 2′-deoxycytidine and follicular development or oocyte quality.

L-alanine is a major amino acid contributor to gluconeogenesis [44]. While in an energy deficit, L-alanine is shuttled from the body to the liver, where it is converted into pyruvate that enters gluconeogenesis to make Glu [45]. During lactation, the use of L-alanine in milk protein synthesis and gluconeogenesis leads to a massive decline in the body fluid of the cow in NEB [46]. This argument aligns with our findings, wherein L-alanine concentration in FF demonstrated a significant decline within the group experiencing NEB conditions. L-alanine is also considered an important noninvasive biomarker for the prediction of oocyte quality before in vitro embryo production [47,48]. The lower concentration of L-alanine in the FF could be the reason for the low fertility in cattle in NEB [49].

Creatine is commonly associated with muscle energy metabolism, while data regarding its role in reproduction is still scanty. It influences ovarian cell metabolism, which regulates follicular development and fertility outcomes [50]. It is present in the FF and may be considered a marker for the oocyte maturity in the follicle [51]. Creatine in conjunction with creatine phosphate constitutes an efficient system that maintains the cellular energy balance by buffering the high-energy phosphate [52]. In follicles exposed to NEB-imitated conditions, we found its downregulation in FF. Although it has been detected in cumulus cells [53], the function of creatine in oocyte development is still unclear [50]. However, an interesting function of creatine metabolism has been observed during pre-fertilization activation of sperm cells [54]. Therefore, oocytes cultured in the NEB environment show slow maturation, fertilization, and oocyte cleavage rate [55]. Based on our conjectures, one possible explanation for this slow development could be low creatine levels, which warrants further investigation.

Trimethylamine-N-oxide (TMAO) is a smaller colorless amine oxide synthesis occurring through the microbial metabolism and enzymatic action of hepatocytes [56,57]. It has a protective role in protein stability during various abiotic stresses, including heat, salinity, hydrostatic and elevated urea concentration in marine animals [56]. TMAO plays a crucial role in maintaining cellular function under elevated osmotic stress conditions [58]. The reduction in osmotic pressure may enhance lipid retention in cattle oocytes [59]. Therefore, elevated TMAO is considered beneficial for fertility in cattle [60]. In our study, the TMAO level in FF was significantly downregulated under NEB. Ji et al. reported that NEB impairs the cumulus cell (CC) structure and the metabolic processes that cause poor oocyte quality [61]. However, a more comprehensive exploration of the role of TMAO-induced osmotic pressure in the structural anomalies observed in the CC or oocytes necessitates further work. Nagy et al. reported that TMAO in the FF negatively correlates with the oocyte quality and fertilization rate in a human study [62]. However, more research is needed to determine the role of TMAO in oocyte development under NEB.

Asparagine is a non-essential amino acid that plays an important role in various metabolic pathways, including the synthesis of Glu, amino acids, proteins, nucleotides and lipids [63]. Additionally, it also maintains the intracellular buffering capacity [47,64]. It is involved in a protein N-linked glycosylation process, in which glycans attach to asparagine. Any defective step in this process may lead to serious congenital disorders impacting female reproductive functions, including oogenesis and folliculogenesis [65]. The depletion of asparagine may enhance the apoptosis rate [66]. Under NEB, as a compensatory mechanism, cattle utilize asparagine as a nitrogen donor for the synthesis of essential proteins and to produce Glu through gluconeogenesis [67]. The depleted asparagine reduces cellular growth and proliferation [63]. In our findings, we observed a significant decline in asparagine levels in the FF of the NEB group. The asparagine deficiency could potentially cause poor oocyte quality and follicle development in cattle [68].

A surge in circulatory BHBA concentration is being observed during the NEB state in cattle [69]. This can be used as a potential biomarker to gauge postpartum fertility. BHBA represents 70% of the total ketone bodies and a strong correlation exists between the serum and FF abundance [24,70]. In the present study, the BHBA was significantly increased in the NEB follicle group. The elevated BHBA concentration negatively affects the follicular microenvironment, oocyte maturation, and overall fertility [63,71]. The pyruvate dehydrogenase energy pathway is disrupted by the elevated BHBA, and intracellular acetylation lowers the oocyte potential [70]. Oocyte deterioration and a decreased conception rate are specifically associated with BHBA-associated elevated oxidative stress, metabolic disruption, and epigenetic modification.

Pantothenic acid (vitamin B5) is an important vitamin in the metabolism of dairy cows [72]. It is an essential component of coenzyme A (CoA), an important factor in carbohydrate, lipid, and protein metabolism [73]. Apart from the metabolic pathways, CoA also acts as an antioxidant in the body [74]. It signifies when cattle are under metabolic stress and mobilizes their fat reserves for energy synthesis. In the current study, we found a higher expression of pantothenic acid in the NEB group. Due to Glu depletion, the follicular cells adopt alternative energy synthesis pathways for follicular and oocyte development. They may utilize BHBA as an alternative energy source during NEB. In mitochondria, BHBA is converted into acetoacetate and aceto-acetyl-CoA. This aceto-acetyl-CoA is, further, the acetyl-CoA that enters the Kreb cycle to produce ATPs [75]. Pantothenic acid is essential for the synthesis of CoA, which further takes part in the conversion of BHBA to energy through the Krebs cycle [76]. Metabolic stress activates peroxisome proliferator-activated receptors (PPARα), leading to modulation of the gene expression promoting lipid metabolism and energy homeostasis. It also upregulates enzymes related to the biosynthesis of pantothenic acid or its metabolites [77]. The higher expression of pantothenic acid in the NEB group FF could be due to starvation-associated gene modulations in the follicular cells.

Many amino acids in FF play an important role in energy generation [78]. They also contribute to maintaining the oxidative balance in the FF required for oocyte maturation [79]. NEB alters the amino acid profile in FF, which may negatively impact follicular development and oocyte competence [80]. A significant decline in the level of L-proline (proline) was observed in the NEB group. Proline was enriched in the arginine proline pathway. Our findings agree with previous studies, which advocated that amino acid and energy synthesis are related changes that are observed in transition cows [59,81,82]. Both arginine and proline are glucogenic amino acids and play their role in meeting the energy deficit. Declined concentrations of such amino acids could be an important marker for detecting metabolic stress in cows [83]. This reduction is associated with the urea cycle and linked to α-ketoglutarate synthesis, whereas α-ketoglutarate is a crucial intermediate in the tricarboxylic cycle [84]. Arginine is metabolized in several compounds, including nitric oxide (NO), polyamines and proline [85]. NO enhances the blood perfusion of the vessels and improves follicular development and oocyte maturation [86,87]. Arginine supplementation may enhance NO production, improving the conception rate [88,89]. Proline can be used for Glu synthesis during starvation; a decline in its level indicates the state of NEB. Proline deprivation may impair collagen production, affecting tissue repair and overall structural integrity [90]. It has been observed that collagen comprises about 12.5% of hydroxylated prolines [91,92]. Studies have revealed the beneficial impact of collagen on oocyte maturation and developmental capacity [93,94,95,96]. Therefore, we speculate that proline downregulation in FF may impair follicular development and oocyte quality in NEB cattle. It has been well established that the arginine proline pathway is enriched in cows under NEB, reflecting their role in metabolic adoption [59,97]. The exclusive pathway is involved in the biosynthesis of crucial molecules, including NO, polyamines, glutamate, creatine, and agmatine, which are essential for different biological functions.

In vitro, follicular culture is a promising tool with immense potential for various fertility-related applications, including oocyte maturation for embryo development and genetic preservation. It may develop a deeper understanding of the complex regulatory mechanisms of follicular development under various physiological, pathological, and drug-testing conditions [98]. Although in vitro models provide insights into the impact of NEB on FF metabolites, translating these findings to in vivo conditions requires careful consideration of the complexities within the living organism. Numerous factors, such as the blood–follicle barrier [99], waste excretion, immune responses, and overall health conditions must be taken into account before comparing the outcomes with in vitro experiments [23].

## 4. Materials and Methods

Unless declared, all the consumed reagents or chemicals in the experiment were procured from the Sigma-Aldrich Chemical Company (St. Louis, MO, USA).

### 4.1. Preparation of Follicular Culture Media

Antibiotic stock solution (ABS), basic follicular culture media and two energy level stock solutions were prepared. Firstly, for ABS, penicillin (10,000 IU/mL) and streptomycin (10,000 IU/mL) were mixed in Dulbecco’s phosphate-buffered saline (DPBS). The homogenized solution was filtered through a 0.22 μm syringe filter. Secondly, the basic follicular culture media solution was prepared using 89% Dulbecco’s modified eagle medium (DMEM), 10% fetal bovine serum (FBS), and 1% ABS [100]. Moreover, positive energy balance (PEB) stock solution was prepared by dissolving 0.1261 g of sodium 3-hydroxybutyrate and 0.684 g of Glu in 10 mL of DPBS solution, stored at 4 °C. Similarly, NEB stock solution was prepared by mixing 0.3028 g of sodium 3-hydroxy butyrate and 0.4684 g of Glu in 10 mL of DPBS solution, stored at 4 °C. The basic follicular culture media stock solution was poured into two 50 mL tubes [101]. The media were centrifuged and subsequently filtered through a 0.22 μL syringe filter. Following this, a volume of 500 μL was removed from each tube, and substituted by an equivalent volume of one of the energy-level stock solutions. The solution containing PEB stock solution fraction was used to simulate a PEB during follicular culture, whereas the solution containing the NEB stock solution fraction was used to mimic in vitro NEB conditions for growing follicles.

### 4.2. Follicle Enucleation and In Vitro Culture

Dairy cattle ovaries were procured from the local slaughterhouse. They were submerged into a prewarmed (37 °C) normal saline-containing insulated bottle and transferred to the laboratory within 2 h. On arrival, the ovaries were rinsed with pre-warmed antibiotics-mixed normal saline, and excessive tissue masses were removed using scissors. Afterwards, the ovaries were washed thrice in normal saline to remove blood stains and other undesired material. Further washed ovaries were transferred to a sterile dissection room for follicular isolation. Enucleation of follicles was performed using sterile instruments (such as scissors, a scalpel blade, and tweezers) on an aseptic working table. Follicles having yellow coloration with diameters ranging from 9 to 11 mm were deemed suitable for subsequent in vitro culture. The ovaries were held with forceps within a Petri dish (90 mm), followed by a precise incision with a scalpel blade to excise the follicle from the ovarian cortex [102]. Post-enucleation, follicles were kept in prewarmed antibiotics containing saline solution. Further, they were washed twice in DPBS at 37 °C and subjected to a final rinse with prewarmed (37 °C) follicle culture media. A 12-well culture plate was used, and half of the wells were labelled as PEB and NEB groups accordingly. The culture media was added to each group well alongside the single follicle. Finally, the cultured plate was placed in an incubator (37 °C and 5% CO_2_) for 24 h.

### 4.3. Follicular Fluid Aspiration and Extraction

After 24-h of incubation, FF samples were aspirated for metabolomic analysis. Briefly, cultured follicles were transferred from the culture plate to another sterile culture dish. With some modifications to the method adopted by Warzych et al., a 1 mL disposable syringe directly aspirated the FF, then stored in 1.5 mL microtubes [103]. The tubes were labelled with the respective well code and all the samples were centrifuged at 3000 rpm/min for 10 min. The supernatant was then carefully collected and shifted into cryovials. The FF samples were snap-frozen and stored at −80 °C till further analysis. The preserved FF samples were thawed at 4 °C. The FF samples from each group were further divided into three subsets to avoid individual variation. They were mixed randomly in their respective groups, and subsets were marked separately as the NEB group: NEB-A, NEB-B and NEB-C and the PEB group: PEB-A, PEB-B and PEB-C). A total of 60 μL of FF sample was taken and mixed with a pre-cooled methanol/acetonitrile/water solution (2:2:1, *v*/*v*). The mixture was vortexed and then subjected to low-temperature sonication for 30 minutes, followed by a 10-min incubation at −20 °C. Further, the samples were again centrifuged at 14,000× *g* for 20 min at 4 °C. The supernatant was collected and vacuum-dried. During mass spectrometry analysis, 100μL of vacuum-dried solution (acetonitrile/water = 1:1, *v*/*v*) was added to redissolve the sample. The mixture was vortexed and centrifuged at 14,000× *g* for 15 min at 4 °C, and the supernatant was collected for metabolite analysis. 

### 4.4. Metabolite Analysis in Follicular Fluid

The non-targeted metabolites were detected utilizing Ultra-High Performance Liquid Chromatography (UHPLC) (Agilent Technologies, Santa Clara, CA, USA). The UHPLC system was connected to a Quadrupole Time-of-Flight Mass Spectrometry (Q-TOF/MS) extension (AB Sciex TripleTOF 6600). The Infinity column measured 2.1 mm × 100 mm, 1.7 μm (Waters Corporation, Wexford, Ireland). The mobile phase was comprised of solvent A (water, and ammonia 1:1) and solvent B (acetonitrile). The gradient elution program was set as 95% Solvent B for 0–0.5 min, a transition from 95 to 65% B over 6.5 min, 65 to 40% B in 1 min, and then maintained for 1 min. Later, a return to 95% in 0.1 min with 3 min re-equilibration was employed accordingly. The Q-TOF/MS apparatus worked in electrospray ionization/ESI mode. The gas temperature of 600 °C, auxiliary heating gas (Gas1) at 60 arbitrary units, and secondary gas (Gas2) at 60 arbitrary units were maintained, whereas the curtain gas pressure was applied as 30 units. The ion spray voltage floating (ISVF) was set as ±5500 V. The mass spectra were recorded in the ranges of 60–1000 Da for MS1 and 25–1000 Da for MS2. Mass spectra were acquired at a rate of 0.20 and 0.05 spectra/s for MS1 and MS2, respectively. The collision energy was adjusted to 35 ± 15 eV for MS2, accompanied by a declustering potential of ±60 V. The mass spectrometry data were subsequently employed for the quantification of target analytes (Figure 8).

### 4.5. Data Analysis

The raw data file underwent conversion to mzXML format via MSConvert tool, ProteoWizard [104]. The converted file was subjected to XCMS software 3.7.0 for peak alignment, retention time correction, and peak area extraction. The metabolites were identified by using an in-house database established with authentic standards. The data were analyzed by the R (ropls) program. The criteria for data analysis included FC > 1.5 or FC < 0.67, *p* < 0.05, and variable importance in projection (VIP) scores > 1. Univariate statistical analysis included fold of variation analysis, Student’s *t*-test/non-parametric test, etc. The KEGG analysis was performed to elucidate the pathways associated with the differentially expressed metabolites.

## 5. Conclusions

It has been concluded that the FF of developing follicles under NEB contains distinct metabolites linked to critical metabolic pathways that are vital for oocyte quality. Those differentially expressed metabolites may serve as potential biomarkers to predict the oocyte quality before procedures such as in vitro fertilization or oocyte cryopreservation. Consequently, additional research is necessitated to explore their ramifications on the enhancement of fertility in both in vivo and in vitro conditions.

## Figures and Tables

**Figure 1 ijms-25-12629-f001:**
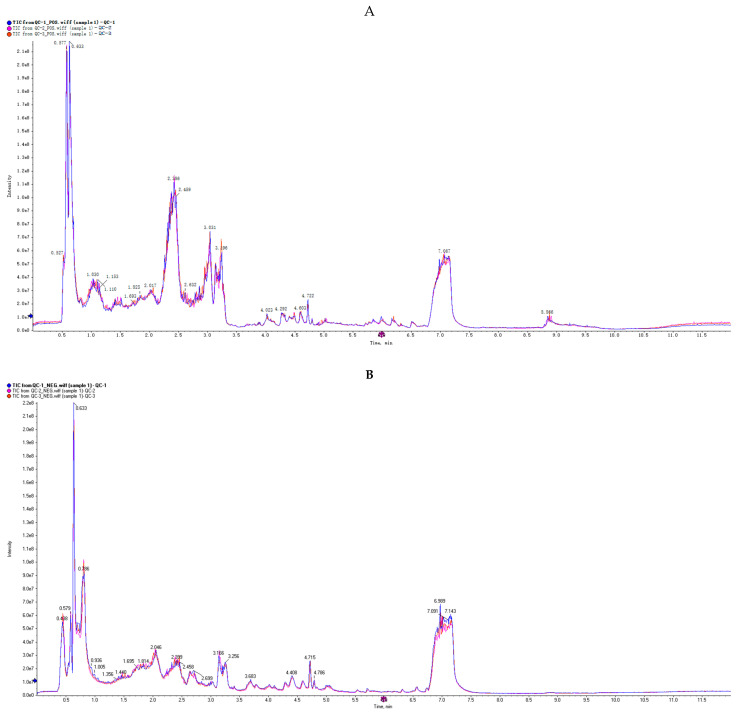
Chromatogram illustrates the total ion current (TIC) with respect to the retention time of metabolite in QC samples both in positive/+ (**A**) and negative (**B**) ions of electrospray ionization (ESI) modes. The signal intensity is displayed on the *y*-axis in million counts, and the retention time is displayed on the *x*-axis in min. The QC1, QC2, and QC3 are represented by the colors blue, pink, and red in the spectrum, respectively.

**Figure 2 ijms-25-12629-f002:**
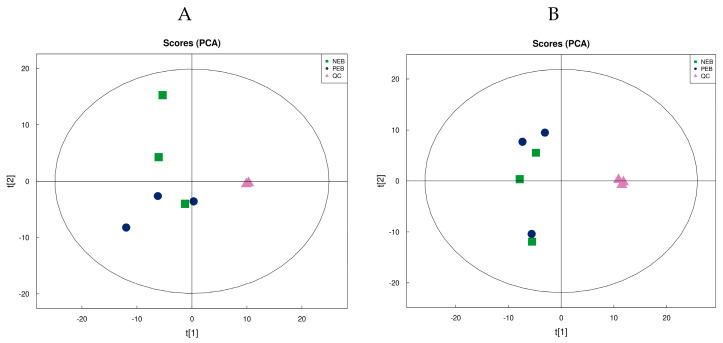
PCA of differential metabolites in follicular fluid from dairy cows under different metabolic states. (**A**) PCA plot illustrates metabolite distribution in positive ion samples. (**B**) PCA plot displaying metabolite profiles from negative ion metabolites. Green squares represent NEB metabolites, blue circles represent PEB metabolites, and pink triangles represent QC. The *x* and *y* axis show the first two principal components (PC1 and PC2) which explain the greatest variance in the data. The label “t[1]” on the *x*-axis represents the first principal component (PC1) of the Principal Component Analysis (PCA).

**Figure 3 ijms-25-12629-f003:**
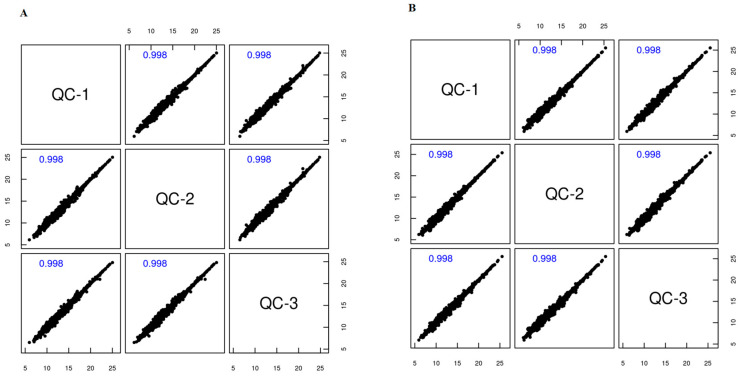
The figure shows correlation profiles of QC samples in metabolomic analysis of FF from dairy cows. (**A**) Correlation matrix for positive ion mode QC samples. (**B**) Correlation matrix for negative ion mode QC samples. Each subplot represents pairwise comparisons between three QC samples (QC-1, QC-2, QC-3), with correlation coefficients shown in blue. The high correlation coefficients (≥0.998) indicate strong reproducibility and stability of the metabolomic measurements across both ionization modes.

**Figure 4 ijms-25-12629-f004:**
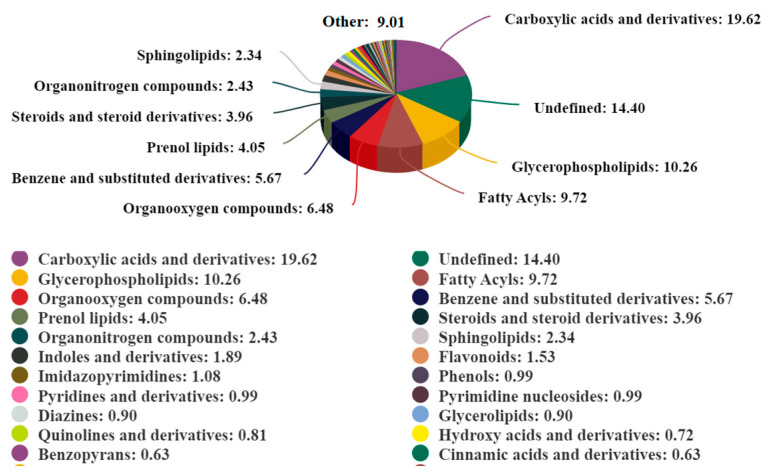
Distribution of metabolite classes (colors) identified in FF of dairy cows. The pie chart illustrates the relative proportions of various metabolite classes, expressed as percentages (%).

**Figure 5 ijms-25-12629-f005:**
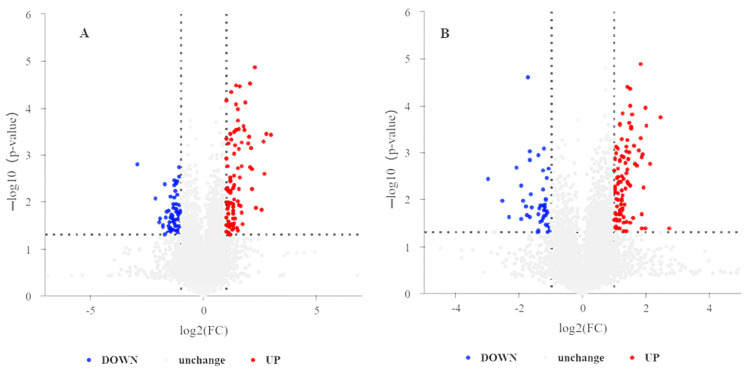
Volcano plots show the differential metabolite expression in FF under NEB and PEB. (**A**) Positive ion mode and (**B**) negative ion mode metabolite profiles. The *x*-axis represents log2 FC. The *y*-axis shows −log10 of *p*-value. Different color dots represent metabolites that were significantly upregulated (red), significantly downregulated (blue), and with no significant change (grey). The fold change (FC) threshold is represented by vertical dotted lines, whereas the significance threshold is represented by the horizontal dotted line (*p*-value).

**Figure 6 ijms-25-12629-f006:**
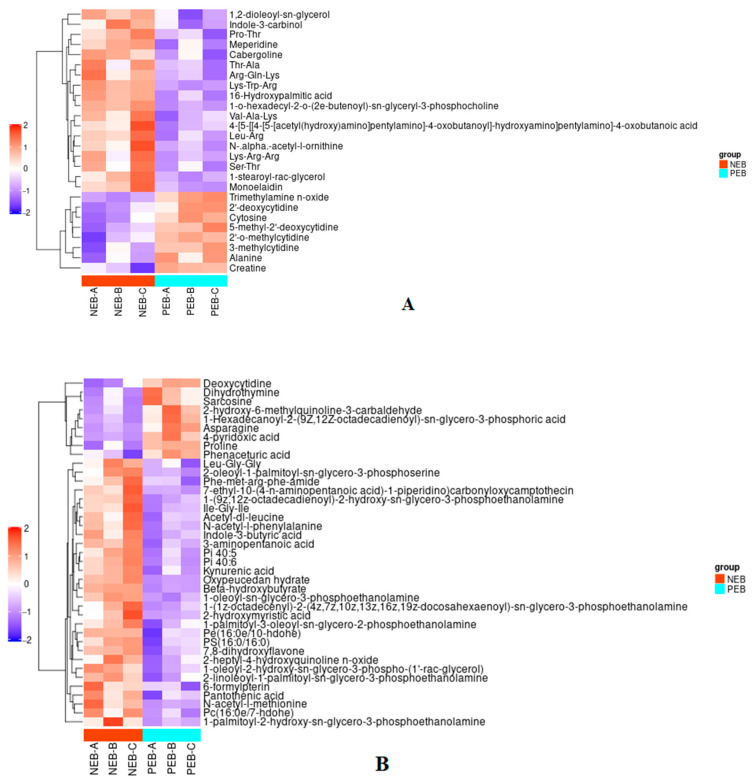
Heatmap showing hierarchical clustering of differentially expressed metabolites in FF NEB and PEB groups. (**A**) Metabolites detected in positive ion mode. (**B**) Metabolites detected in negative ion mode. The color scale denotes the relative abundance of metabolites, with red indicating higher levels and blue indicating lower levels. Each horizontal row represents a distinct metabolite, and each vertical column represents a sample (NEB-A, NEB-B, NEB-C, PEB-A, PEB-B, PEB-C). The dendrograms on the left-hand side represent the hierarchical clustering of metabolites based on their expression patterns.

**Figure 7 ijms-25-12629-f007:**
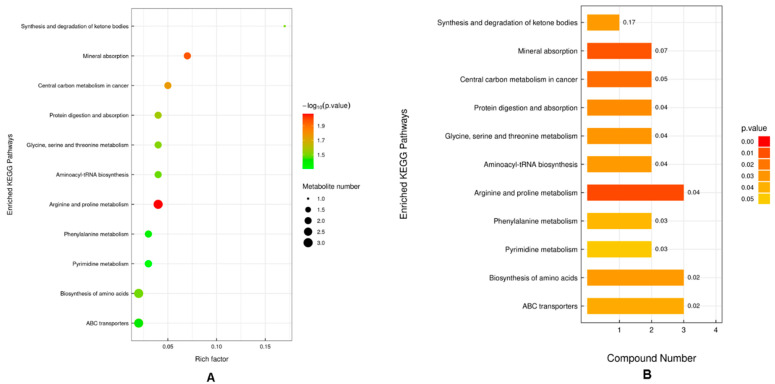
KEGG pathway enrichment analysis of differentially expressed metabolites in FF of dairy cattle under varying energy balance conditions. (**A**) The bubble plot represents the enriched pathways. The *x*-axis gauges the rich factor (ratio of differentially expressed metabolites in a pathway to the total metabolites in that pathway). The *y*-axis enlists the enriched KEGG pathways. The bubble size indicates the number of metabolites, and the color represents the significance level (−log10 (*p*-value)). (**B**) Bar plot of the same enriched pathways, where the *x*-axis shows the number of compounds involved in each pathway, and bars are colored according to *p*-value significance.

**Figure 8 ijms-25-12629-f008:**
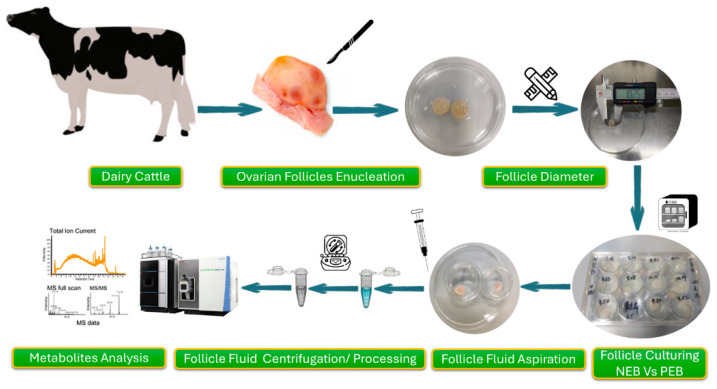
Schematic diagram of the experiment illustrating follicle enucleation, diameter measurement, culturing, FF collection, and metabolite analysis.

**Table 1 ijms-25-12629-t001:** Metabolite quantity in the positive and negative ions for electrospray ionization (ESI) modes.

Detection Mode	Number of Metabolites
Positive Ion Mode	616
Negative Ion Mode	495

**Table 2 ijms-25-12629-t002:** Differential FF metabolites in positive ion mode.

Metabolite Name	*m*/*z*	Rt/min	VIP Value	*p*-Value	Variation
16-Hydroxypalmitic acid	272.26	107.44	1.61	0.00	↑
2′-Deoxycytidine	455.19	208.56	4.35	0.02	↓
5-Methyl-2′-deoxycytidine	126.07	200.14	1.54	0.01	↓
L-Alanine	90.05	351.52	2.24	0.04	↓
Cabergoline	452.31	191.68	2.04	0.04	↑
Creatine	132.08	353.30	10.08	0.03	↓
Cytosine	112.05	208.69	3.01	0.02	↓
Pethidine	174.12	522.11	4.21	0.03	↑
N-Acetylornithine	157.10	328.50	1.28	0.04	↑
Trimethylamine-N-oxide	76.08	337.68	1.02	0.00	↓
1-O-hexadecyl-2-o-(2e-butenoyl)-sn-glyceryl-3-phosphocholine	550.38	190.40	1.96	0.00	↑
Monoglycerides stearate	341.30	33.65	1.64	0.01	↑
1, 2-dioleoyl-sn-glycerol	643.52	32.13	1.11	0.02	↑
2′-Methoxycytidine	258.11	175.28	1.53	0.03	↓
3-Methylcytidine	258.11	252.94	1.06	0.04	↓
4-[5-[[4-[5-[Acetyl(hydroxy)amino] pentyl amino]-4-Oxobutanoyl]-hydroxyamino] pentyl amino]-4-oxobutanoic acid	478.29	199.60	1.05	0.05	↑
Arg-Gln-Lys	216.13	362.92	1.02	0.02	↑
3-Indolemethanol	130.06	124.97	1.50	0.04	↑
Leu-Arg	288.19	396.77	1.26	0.02	↑
Light-Angry-Angry	230.15	363.53	3.82	0.04	↑
Lys-Trp-Arg	245.15	376.42	2.07	0.00	↑
Glyceryl reflex oleate	339.29	32.41	1.66	0.01	↑
Proline-threonine	217.13	374.34	2.56	0.02	↑
Serine threonine	207.11	234.19	1.69	0.05	↑
Thr-Ala	191.08	257.72	2.45	0.05	↑
Val-Ala-Lys	159.11	270.80	3.58	0.03	↑

*m*/*z*: Mass-to-charge ratio; RT: Retention time; min; VIP: Variable importance in projection; (↓): Downregulation; (↑): Upregulation.

**Table 3 ijms-25-12629-t003:** Differential FF metabolites in negative ion mode.

Metabolite Name	*m*/*z*	Rt/min	VIP Value	*p*-Value	Variation
2-Hepyl-4-hydroxyquinoline-N-oxide	144.04	70.14	1.98	0.05	↑
4-Pyridoxic acid	182.04	43.60	9.08	0.00	↓
Asparagine	131.05	308.55	2.28	0.01	↓
β-Hydroxybutyric acid	103.04	228.40	9.91	0.00	↑
Deoxycytidine	226.08	209.31	1.79	0.02	↓
5, 6-Dihydrothymine	165.00	284.55	1.37	0.04	↓
Indole-3-butyric acid	158.08	205.74	2.29	0.02	↑
4-Hydroxyquinoline-2-carboxylic acid	188.03	61.82	1.40	0.02	↑
N-Acetyl-L-methionine	190.05	194.00	1.45	0.01	↑
N-Acetyl-L-phenylalanine	206.08	187.84	1.93	0.02	↑
Pantothenic acid	437.21	266.21	2.37	0.03	↑
Phenylacetyl glycine	192.05	146.47	17.06	0.04	↓
L-Proline	114.06	316.26	2.55	0.02	↓
Sarcosine	88.04	349.33	2.15	0.04	↓
1-(1z-octadecenyl)-2-(4z,7z,10z,13z,16z,19z-docosahexaenoyl)-sn-glycero-3-phosphoethanolamine	774.53	40.04	1.32	0.03	↑
1-(9z,12z-octadecadienoyl)-2-hydroxy-sn-glycero-3-phosphoethanolamine	476.27	200.75	1.81	0.01	↑
1-Palmitoyl-2-linoleoyl-sn-glycero-3-phosphate	671.46	205.70	1.46	0.01	↓
1-Oleoyl-2-hydroxy-sn-glycero-3-phospho-(1′-rac-glycerol)	509.28	165.39	1.46	0.01	↑
1-Oleoyl-sn-glycero-3-phosphocholine; oleoyl-Lys phosphatidylcholine	478.29	199.14	3.96	0.00	↑
1-Palmitoyl-sn-glycero-3-phosphoethanolamine	452.27	202.03	1.96	0.05	↑
1-Palmitoyl-3-oleoyl-sn-glycero-2-phosphoethanolamine	716.52	41.63	1.16	0.04	↑
2-Hydroxy-6-methylquinoline-3-carboxaldehyde	186.04	221.53	2.25	0.03	↓
2-Hydroxytetradecanoic acid	243.19	78.61	1.70	0.04	↑
2-Linoleoyl-1-palmitoyl-sn-glycero-3-phosphoethanolamine	714.50	153.31	2.20	0.03	↑
2-Oleoyl-1-palmittin tin glycerol-3-phosphocholine	760.51	194.73	1.62	0.02	↑
3-Aminvaleric acid	116.07	205.55	1.56	0.01	↑
6-Formrexate	190.04	294.33	3.66	0.04	↑
7-Ethyl-10-(4-n-amino pentanoic acid)-1-piperidino)carbonyloxycamptothecin	617.27	173.51	2.25	0.02	↑
7, 8-Dihydroxyflavonoids	253.11	247.03	1.05	0.01	↑
Acetyl leucine	172.10	193.37	3.06	0.04	↑
With-Gly-With	300.19	396.26	3.22	0.03	↑
L-Leucyl glycylglycine	244.13	464.19	1.00	0.05	↑
Hydration of oxidized imperatorin	349.08	228.38	4.10	0.00	↑
Pc (16:0e/7-hdohe)	866.58	41.63	1.03	0.02	↑
Bei (16:0e/10am-10am)	764.52	148.18	2.97	0.03	↑
Phe-met-arg-phe-amide	597.30	242.68	1.76	0.04	↑
Pi 40:5	911.56	189.57	1.85	0.02	↑
Pi 40:6	909.54	188.98	1.25	0.01	↑
PS (16:0/16:0)	734.51	146.27	1.26	0.03	↑

*m*/*z*: Mass-to-charge ratio; RT: Retention time; min; VIP: Variable importance in projection; (↓): Downregulation; (↑): Upregulation.

## Data Availability

All the data supporting the conclusions in this article have been presented in the manuscript.

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
