# Peer review of "Unravelling the Signature Follicular Fluid Metabolites in Dairy Cattle Follicles Growing Under Negative Energy Balance: An In Vitro Approach"

_ijms, 2024, doi:10.3390/ijms252312629_

Round 1

Reviewer 1 Report

Comments and Suggestions for Authors

Shahzad et used the in-vitro approach to investigate the signature follicular fluid metabolites in dairy cattle follicles growing under negative energy balance. Overall the design is rational and results are interesting in the field of dairy. I have a few comments that I want to highlight.

1. Figure 1 legends can be explained little bit more such as describing the peaks.

2. Methods of identification and classification of metabolites need more explanation.

3. I am wondering there is no figure or explanation about GO (Gene Ontology) terms and GO enrichment statistics.

4. It would be better if authors validate some of findings through qPCR or WB.

Author Response

  1. Figure 1 legends can be explained little bit more such as describing the peaks.

Response 1- Needful done (Bold in the designated section)

  1. Methods of identification and classification of metabolites need more explanation.

Response 2- Needful done (Bold in the designated section)

  1. I am wondering there is no figure or explanation about GO (Gene Ontology) terms and GO enrichment statistics.

Response 3- Since our presented study was primarily focused on identifying differential metabolites using LC MS-MS and their classification by the KEGG database for the identification of metabolites. Consequently, our findings are predominantly concerned with metabolite profiling rather than the expression of genes. Hence, Gene Ontology (GO) terms and GO enrichment analyses were not incorporated, as they are predominantly associated with genomic studies. We assert that, in alignment with our objectives of discerning differential metabolites, KEGG-based classification effectively fulfills the intended purpose.

  1. It would be better if authors validated some of findings through qPCR or WB.

Response 4- I express my profound gratitude for your insightful suggestion, which we shall incorporate into our forthcoming experimental endeavors. Regarding the study presented, our principal aim was to elucidate the metabolites utilizing LC MS-MS and systematically categorize them; therefore, we adhered strictly to our objectives up to this juncture. I sincerely appreciate your thoughtful guidance once more.

Reviewer 2 Report

Comments and Suggestions for Authors

In the submitted manuscript, the authors have identified key follicular fluid metabolites in dairy cattle follicles that grow under negative energy balance, using an in vitro approach. Overall, the manuscript is well-written. The conclusions are supported by the data and the discussion references relevant work in the field while emphasizing the significance of the findings. The figures are clear and informative. I would suggest that the authors should include the limitations of the study.

Author Response

Comment In the submitted manuscript, the authors have identified key follicular fluid metabolites in dairy cattle follicles that grow under negative energy balance, using an in vitro approach. Overall, the manuscript is well-written. The conclusions are supported by the data and the discussion references relevant work in the field while emphasizing the significance of the findings. The figures are clear and informative. I would suggest that the authors should include the limitations of the study.

Response Needful Done 

Reviewer 3 Report

Comments and Suggestions for Authors

The introduction provides sufficient information including relevant references.

I consider that the material and methods section is very brief and could be expanded further by explaining in detail the methods employed. 

Any comments on the ability of the in vitro methodology employed to reproduce the biological phenomena under study?    I think it would be appropriate to include this as part of the discussion.

It is not clear how the high correlation coefficients indicate strong reproducibility and stability of metabolomics measurements across both ionisation modes. I believe this needs to be explained further.

The conclusions are written as results, so it is suggested to improve this part of the paper.

Comments on the Quality of English Language

The paper needs to be improved in some sections as follows

The introduction provides sufficient information including relevant references.

I consider that the material and methods section is very brief and could be expanded further by explaining in detail the methods employed. 

Any comments on the ability of the in vitro methodology employed to reproduce the biological phenomena under study?    I think it would be appropriate to include this as part of the discussion.

It is not clear how the high correlation coefficients indicate strong reproducibility and stability of metabolomics measurements across both ionisation modes. I believe this needs to be explained further.

The conclusions are written as results, so it is suggested to improve this part of the paper.

Author Response

Comment 1. The introduction provides sufficient information including relevant references.

Response 1: Thank you for your kind appreciation. It is a token of encouragement for us

Comment 2. I consider that the material and methods section is very brief and could be expanded further by explaining in detail the methods employed. 

Response 2: Thank you for indicating this. Needful done where needed, please

Comment 3. Any comments on the ability of the in vitro methodology employed to reproduce the biological phenomena under study? I think it would be appropriate to include this as part of the discussion.

Response 3: Very nice idea. We have added that in the discussion section

Comment 4. It is not clear how the high correlation coefficients indicate strong reproducibility and stability of metabolomics measurements across both ionisation modes. I believe this needs to be explained further.

Response 4: In our study, the correlation coefficient among the QC samples exceeded 0.9, indicating a high reproducibility level in the experimental procedure and affirming the reliability of the subjected data. Generally, a correlation coefficient reading closer to 1 or -1 indicates a strong correlation. Therefore, in scientific experiments, a high correlation coefficient like 0.9 or above is highly appreciated which is an indicator that results are consistent and reliable across samples, providing greater confidence in the data.

Comment 5: The conclusions are written as results, so it is suggested to improve this part of the paper.

Response 5: Thank you for your suggestion, we have done so.

In the final reckoning, we all are humbly thankful to you for such professional and generous feedback to polish our manuscript.